# Exploratory Study on DNA Barcode Combined with PCR-HRM Technology for Rapid and Accurate Identification of Necrophilous Fly Species

**DOI:** 10.3390/insects16060590

**Published:** 2025-06-04

**Authors:** Bo Wang, Shan Ha, Jifeng Cai, Yixin Ma, Dianxin Li, Jianhua Chen, Jianqiang Deng

**Affiliations:** 1Hainan Provincial Tropical Forensic Engineering Research Center & Hainan Provincial Academician Workstation (Tropical Forensic Medicine), Key Laboratory of Tropical Translational Medicine of Ministry of Education, School of Basic Medicine and Life Sciences, Hainan Medical University, Haikou 571199, China; sapphirestrings@sina.com (B.W.); 1186527967hs@sina.com (S.H.); xmbebrave@163.com (Y.M.); 15542672385@163.com (D.L.); 2Department of Forensic Medicine, Key Laboratory of Forensic Medicine, School of Basic Medicine, Xinjiang Medical University, Urumqi 830017, China; cjf_jifeng@163.com; 3School of Public Health, Hainan Medical University, Haikou 571199, China; 4Department of Forensic Science, Xiangya School of Basic Medical Sciences, Central South University, Changsha 410013, China

**Keywords:** forensic entomology, necrophilous flies, HRM, species identification

## Abstract

Determining the time of death is critical in solving crimes, but traditional methods struggle to determine this beyond 24 h. This study developed a DNA-based technique (high-resolution melting—HRM) to rapidly identify necrophagous flies which colonize corpses and help estimate the time of death. Focusing on 10 species of flies from Hainan Island, we designed a new genetic marker (COX2-519/COX2-615 primers) to distinguish species by analyzing tiny DNA variations. This marker successfully identified all 10 species of flies, including larvae, even in degraded samples stored for over two years. The method is faster (2–3 h) and more cost-effective than sequencing, and works in a single step, reducing contamination risks. At the same time, it avoids the subjective human influence of morphological identification and is more accurate. By enabling precise species identification, this tool helps forensic experts estimate death timelines more accurately, aiding criminal investigations and justice. This study highlights the potential of this marker as a reliable standard for forensic entomology, particularly in regions with limited access to advanced DNA sequencing.

## 1. Introduction

Estimating the time of death has always been one of the most critical and challenging issues in judicial practice. In forensic medicine, determining the postmortem interval (PMI) is crucial for estimating time since death, which can aid in narrowing investigative timelines and further focus investigative efforts [1]. Traditional methods for estimating the time of death are primarily based on early postmortem changes, which are generally only effective for estimating the PMI within 24 h after death [2,3]. For longer PMIs, there is no widely accepted and reliable method. Currently, the forensic community believes that forensic entomology holds promise as an effective approach for estimating longer PMIs [4].

Forensic entomology refers to the application of entomological theories and methods to study and solve forensic problems related to insects. Its primary research focus is on necrophagous insects, which are insects that feed on decaying organic matter [5]. Most of these studies have focused on readily available necrophilic flies. After the death of an animal or human, adult necrophagous flies are among the first to arrive at the corpse to lay eggs or larvae. The larvae feed on the decomposing tissue, mature, and eventually pupate, with the pupae later developing into adult flies. The life cycle and developmental stages of these flies are highly regular [6]. Over time, different species of necrophagous flies arrive at the corpse in a predictable succession pattern. By analyzing this succession pattern, it is possible to estimate the PMI in forensic investigations. While traditional methods based on postmortem changes are only effective within 24 h after death, forensic entomology can extend this estimation to 2 days to 2 weeks, or even several years after death [7]. This is particularly important in the hot and humid tropical regions where this study was conducted, as corpses decompose rapidly, making it difficult to estimate the time of death using conventional methods [8]. As a result, forensic entomology is highly regarded as a promising solution in such environments.

The application of forensic entomology in forensic practice relies on the accurate identification of necrophagous fly species. Morphological and molecular genetic methods are the two main approaches for species identification. However, morphological identification requires extensive expertise and experience, which limits its application in forensic practice. Over the last two decades, with the rapid development of molecular biology, particularly the advancement of species classification technologies based on the analysis of genetic material such as DNA, molecular species identification techniques have begun to play an increasingly important role in forensic entomology [9,10,11]. Currently, databases established based on mitochondrial DNA (mtDNA) studies of flies have enabled the identification of species across 31 orders of insects [12]. Among these, the mitochondrial cytochrome oxidase (COX) gene has been proven to be the most suitable DNA barcoding gene for molecular species identification in forensic entomology, demonstrating significant potential for application. Exploring more sensitive, user-friendly, and rapid DNA analysis techniques remains a critical research direction in forensic entomology [13,14,15].

High-resolution melting (HRM) technology is an emerging analytical technique that is based on single-nucleotide polymorphisms (SNPs) or minor variations in DNA sequences. During the PCR amplification process, differences in melting temperatures generate distinct melting curve profiles, enabling the analysis of genetic variations across different samples [16,17]. HRM technology is characterized by high temperature uniformity, high resolution, and high sensitivity [18], and has been widely applied in various fields for the analysis of DNA sequence differences [19].

Theoretically, HRM can detect sequence variations as small as a single base pair, making it highly suitable for species identification studies, such as SNP scanning, genotyping, and species identification, which are essential in forensic practice. The HRM process can be performed in a single tube, minimizing the risk of contamination; additionally, it is simple, fast, and accurate, making it highly suitable for forensic applications [20,21,22,23]. HRM technology has been successfully applied in forensic identification. In forensic evidence, HRM enables rapid detection of mixed DNA samples [24,25] (Smith et al., 2024; Torres et al., 2023), saliva stain typing [26] (Yang et al., 2022), and male DNA identification [27] (Ginart et al., 2024). It also facilitates body fluid differentiation through methylation analysis [28,29] (Antunes et al., 2021; Fujimoto et al., 2021) and distinguishes monozygotic twins [30] (Romanos et al., 2021). In toxicology, HRM aids in identifying hallucinogenic mushrooms [31] (Zhang et al., 2021) and optimizing drug detection in hair samples [32,33] (Spear et al., 2022). For species identification, HRM effectively differentiates ivory sources [34] (Carrothers et al., 2021), mammalian species [35] (Kriangwanich et al., 2021), and toxic plants [36] (Anthoons et al., 2022). Additionally, HRM combined with machine learning [37] (Ozkok et al., 2021) and statistical methods [38] (Cloudy et al., 2024) enhances data analysis capabilities, demonstrating its efficiency and versatility in forensic applications.

However, there are relatively few reports on the application of HRM in forensic entomology. In 2023, our research team attempted to apply HRM technology for species identification of necrophagous flies, but the coverage of species and amplification efficiency still require improvement [39]. Therefore, selecting appropriate DNA barcoding regions for HRM and establishing corresponding molecular species identification protocols for forensic insects are critical steps that must be addressed to enable the widespread application of this technology in forensic practice.

Based on this, we selected dominant necrophagous fly species collected from Hainan Island in 2021 as the research subject. DNA was extracted from these flies, and candidate DNA barcoding primers were screened from published studies. Additionally, new mitochondrial DNA barcoding primers were designed using software. We then developed a technical HRM protocol to explore and establish a reliable and effective HRM-based species identification system for forensic entomology. The stability and feasibility of this system were validated, and it was further applied to explore the species identification of fly larvae.

## 2. Materials and Methods

### 2.1. Sample Selection and DNA Extraction

To ensure the practical forensic value of this study, we selected insect samples that had been preserved for a long time, which are more challenging for molecular species identification and are commonly encountered in forensic case scenarios, as the research subjects. The necrophilous fly samples collected in 2021 were stored in centrifuge tubes at room temperature under dry conditions to simulate natural DNA degradation. The fly samples were morphologically classified using the “Key to Common Flies of China [40]” and “*Necrophilous Flies of China* [41]”. The DNeasy^®^ Blood & Tissue Kit (QIAGEN, Hilden, Germany) was applied for the extraction of DNA from fly samples, and the method was performed as detailed in [42]. The extracted DNA samples were sequenced to confirm species identification. The samples included three species from the Calliphoridae family (*Chrysomya megacephala (Fabricius, 1794)*, *Hemipyrellia ligurriens (Wiedemann, 1830)*, and *Achoetandrus rufifacies (Macquart, 1843)*), six species from the Sarcophagidae family (*Boettcherisca peregrina (Robineau-Desvoidy, 1830)*, *Parasarcophaga dux (Thomson, 1868)*, *Parasarcophaga misera (Walker, 1849)*, *Parasarcophaga sericea (Walker, 1852)*, *Parasarcophaga ruficornis (Fabricius, 1794)*, and *Seniorwhitea princeps (Wiedemann, 1830)*), and one species from the Muscidae family (*Musca domestica (Linnaeus, 1758)*), totaling ten species as experimental subjects. While testing, we incorporated supplementary fly species samples. Refer to the results for the specific species analyzed.

The concentration of all extracted DNA samples was measured to determine the amount of template DNA to be added. DNA concentration was measured using the Qubit 3.0 Fluorometer (Thermo Fisher Scientific, Waltham, MA, USA) with the Qubit™ 1 × ds DNA HS Assay Kit.

### 2.2. Primer Design

The mitochondrial COXⅠ and COXII genes of the ten necrophilous fly species were searched in the NCBI database, and the complete sequences were saved in FASTA format. Only eight species’ sequences were successfully obtained from the database, and their accession numbers are listed in Table 1. Sequence alignment was performed using ClustalX 1.83 software to identify highly variable regions within the DNA fragments. Primer Premier 5 was used for primer design. For the eight species with complete COXⅠ and COXII sequences available in the database, primers were designed separately. The primer size was set to 20 ± 2 bp, and the PCR product length was limited to 300 bp. High-scoring primers were selected, and those with hairpin structures, mismatches, or self-dimers were excluded to reduce the probability of mismatches. Figure 1 illustrates the primer design process using the COXII gene sequence of *P. misera*.

The specificity of the designed primers was verified using the Primer-Blast tool in the NCBI database. In the “Database” option, the “nr” database was selected. For the “Organism” option, in addition to the default “Homo sapiens”, “Diptera (taxid: 7147)” was added to focus on the order of flies. The Primer GC content (%) settings were adjusted by changing the minimum value from 20.0 to 40.0 and the maximum value from 80.0 to 60.0 to ensure that the primers had an appropriate melting temperature and to prevent the generation of non-specific peaks [43]. All other options were kept at their default settings. Based on the BLAST results, primers capable of amplifying multiple necrophilous fly species were selected, synthesized, and used in subsequent PCR-HRM experiments.

### 2.3. PCR-HRM Analysis

PCR amplification and HRM analysis were performed using the Rotor-Gene Q Real-Time PCR System (Qiagen, Hilden, Germany). Each sample was analyzed in triplicate, and negative controls were included to rule out contamination.

The PCR cycling parameters (Table 2) were as follows: 95 °C for 10 min; 95 °C for 30 s, Tm for 30 s, and 72 °C for 45 s, for a total of 35 cycles; and 72 °C for 10 min. Subsequently, HRM assays were carried out as follows: 95 °C for 1 min and 40 °C for 1 min; the melting temperature was gradually increased from 65 °C at 0.1 °C/s to 90 °C. The PCR reaction system is shown in Table 3.

### 2.4. Verification of the Identification System’s Reliability

A good melting curve result should exhibit a characteristic peak at the melting temperature (Tm) of the DNA [44]. HRM technology can further distinguish the melting temperatures of different DNA sequences with high precision, meaning that each species has a unique melting temperature [45]. This serves as the criterion for evaluating the identification capability of the primer system. We conducted the following experiments to verify the reliability of the system.

(1)Repeatability Test

To assess the stability of the experimental method, DNA was extracted from samples of the same species using the aforementioned protocol, and the experiment was repeated to confirm the reproducibility of the results.

(2)Identification Capability Test

DNA samples with previously confirmed classification capabilities were used as references. Randomly captured samples were amplified simultaneously with the reference DNA; the consistency of the results was observed.

(3)Special Sample Experiment

Flesh flies (Sarcophagidae) were randomly captured and reared to obtain larval samples. DNA was extracted from these larvae, and PCR amplification and HRM analysis were performed simultaneously, using previously classified flesh fly DNA samples as references. This was carried out to verify whether larval samples could be accurately identified using this method.

### 2.5. DNA Sequencing

The PCR-HRM products from the newly designed primers capable of species identification were subjected to Sanger sequencing. The sequencing data were then aligned using ClustalW in the MEGA-X 10.2.2 software to determine the DNA polymorphism. Subsequently, the sequencing data were compared with sequences in the NCBI database to confirm the target gene and the corresponding species.

## 3. Results

### 3.1. Determination of DNA Concentration in Necrophilic Fly Samples

Samples of ten necrophilic fly species were selected for the experiment in 2021; after two years, the double-stranded DNA was not completely broken. Among these samples, the double-stranded DNA concentration of *M. domestica*, *P. dux*, and brown bearded linseed fly was more than 100 ng/µL. Among them, the species with the least amount of DNA was the short-horned linseed fly, at only 3.83 ng/µL; however, the PCR requires a final template DNA concentration of no less than 10 ng. As such, 3 µL of short-horned flax fly template DNA and 4 µL of ddH^2^O were added, and the rest of the template DNA was 2 µL (Table 4).

### 3.2. Primer Selection Conditions

The primary objective of primer selection in this study was to extend the sample identification timeframe and cover a broader range of necrophilous fly species. The selected HRM primers must meet the fundamental requirements for fly species identification, ensuring that the amplification targets include both conserved regions and variable regions. The conserved regions enable the specific amplification of necrophilous flies, while the variable regions allow for differentiation between species.

In addition to meeting basic amplification requirements, the amplified product size should be less than 300 bp. Longer products are prone to mutations during PCR, and the high resolution of HRM can distinguish even single-nucleotide polymorphisms (SNPs) from the original sequence, resulting in distinct melting peaks. Furthermore, the samples used in this experiment had been stored for an extended period, leading to varying degrees of DNA degradation. As a result, longer target genes were difficult to amplify, necessitating the selection of shorter target regions. Table 5 lists all the primers newly designed and obtained from the literature and investigated in this study.

### 3.3. Results of the Species Identification of Necrophilous Fly Species HRM

After repeated testing, one pair of primers from the literature and one pair of newly designed primers were selected for use. The remaining primers either failed to amplify or produced multiple peaks in the HRM results. Detailed information is provided in Table 6.

#### 3.3.1. C1-J-2495/C1-N-2800 Primer HRM Results

The C1-J-2495/C1-N-2800 primers amplified a total of six species, with melting curve peaks for *H. ligurriens* at 76.53, *B. peregrina* at 76.77, *P. dux* at 75.97, *M. domestica* at 75.22, and *Parasarcophaga scopariiformis (Senior-White, 1927)* at 75.90. Three peaks for *S. princeps* existed at 74.75, 77.65, and 80.32. The positions of the other two peaks did not overlap with the other peaks, and were not contaminated by other flies (Figure 2).

#### 3.3.2. COX2-519/COX2-615 Primer HRM Results

The primers, designed based on the mitochondrial COXII gene sequence of *P. misera*, were amplified with yielding single, species-specific melt peaks for all 10 species. The following peaks were observed: *C. megacephala* at 76.22, *H. ligurriens* at 76.70, *A. rufifacies* at 76.45, *B. peregrina* at 76.73, *P. dux* at 77.07, *P. misera* at 76.32, *P. sericea* at 77.20, *P. ruficornis* at 76.52, *S. princeps* at 76.88, and *M. domestica* at 75.70. Of these, only the Tm values of *H. ligurriens* and *B. peregrina* were close, differing by only 0.03 °C (Figure 3).

#### 3.3.3. Other Primer HRM Results

The primers obtained through literature screening, as well as the self-designed primers for HRM experiments, none—except for the above two primers—obtained better results. The rest of the primers mostly exhibited poor amplification, no amplification, or only a certain fly species produced a peak on the melting curve. Another situation is that the melting curve of the amplicon has multiple peaks, as shown in Figure 4 below, whereby the primer is COX2-365/COX2-521, the length of the amplified fragments is 157 bp, and the single peaks of the HRM experiment of the amplicon of this primer are *C. megacephala*, *A. rufifacies*, *H. ligurriens*, and *M. domestica*. Meanwhile, *B. peregrina*, *S. princeps*, *P. misera*, *P. sericea*, and *Leucomyia cinerea (Fabricius, 1974)* all showed multiple peaks.

### 3.4. Reliability of the Identification System

#### 3.4.1. Repeatability of the Experiment Results

To validate the amplification accuracy of the COX2-519/COX2-615 primers, *B. peregrina*, *P. dux*, *P. misera*, *P. sericea*, and *P. ruficornis*—five species of Sarcophaga—were selected, re-extracted the DNA and the experiment was repeated to determine the recognition ability (Figure 5). The peak melting curves for each fly species are seen for *B. peregrina* at 76.83, *P. dux* at 77.17, *P. misera* at 76.35, *P. sericea* at 77.22, and *P. ruficornis* at 76.60, which differed slightly from the previous results in terms of peak value, all of which were slightly larger than the previous group, with an error range of 0.02 to 0.1; the peak heights also differed considerably.

#### 3.4.2. Species Identification Competency Test Results

A necrophilic fly was randomly captured to extract DNA, and the primers COX2-519/COX2-615 were applied to perform PCR-HRM experiments simultaneously with the DNA of the ten fly species that could be identified in the previous section, as shown in Figure 6. The yielded single, species-specific melt peaks of the fresh sample were higher, but the Tm value of the melting curve was consistent with that of *B. peregrina*, both of which were 76.73. Therefore, the unknown fly species could be *B. peregrina*.

A necrophilic fly was also randomly captured, its DNA was extracted, and PCR-HRM experiments were applied with C1-J-2495/C1-N-2800 primers, as shown in Figure 7. It was observed that the Tm value of this sample was 74.70, which differed only by 0.05 °C from the first peak of *S. princeps*; however, only a single peak was present in this sample.

#### 3.4.3. Larval Sample Validation

The captured Sarcophaga used for rearing and obtaining samples of the larvae produced, as well as DNA extraction, were used for PCR-HRM experiments, together with the re-extracted DNA samples of five species of Sarcophaga flies; COX2-519/COX2-615 primers were applied with yielded single, species-specific melt peaks, as shown in Figure 8. The values of Tm were 75.26 for *B. peregrina*, 77.40 for *P. dux*, 77.20 for *S. princeps*, 76.96 for *P. misera*, 77.14 for *P. sericea*, and the Tm value of the larval sample to be tested was 77.50, which differed from *P. dux* by only 0.1 °C. The sample of the unknown fly larvae was considered as a possible *P. dux*.

### 3.5. Sequencing Results

The amplification products of the newly designed COX2-519/COX2-615 primers after the PCR-HRM test were subjected to one-generation sequencing for a total of ten samples; the sequencing results are shown in Figure 9. The sequences of all ten samples were different, the sequencing lengths obtained ranged from 76 bp to 96 bp, and the site possessed both length polymorphism and sequence polymorphism.

When we look up the results of sequencing in NCBI-BLAST, there are two situations. The first is that for species such as *C. megacephala* (Figure 10A), *P. dux* (Figure 10B), etc., although 100% identical sequences could not be found, the results of the search and the experimental results are only 1 bp apart, and the “sort by percent identity” is greater than 90%. The other situation is the relatively less studied species such as *S. princeps* (Figure 10C), where the “sort by percent identity” is less than 90% and no identical species exist in the search results.

## 4. Discussion

The melting curve is generated as the double-stranded DNA (dsDNA) dissociates into single strands, causing the saturation dye to separate, resulting in a change in absorbance. This forms a curve where absorbance varies with temperature. Since different DNA sequences have distinct melting temperature (Tm) values, characteristic peaks are formed at these temperatures, enabling genetic analysis. The melting temperature of DNA is primarily influenced by the following three factors: DNA length, G-C content, and base sequence [48,49].

For example, if a DNA sequence lacks one G-C pair (equivalent to losing three hydrogen bonds), the change in melting temperature (ΔTm) is approximately 6 °C.

If a G-C pair is mutated to an A-T pair (resulting in the loss of one hydrogen bond), the ΔTm is approximately 0.6 °C.

If the number of hydrogen bonds remains unchanged (e.g., A-T to T-A or G-C to C-G mutations), the ΔTm of the PCR amplification products is approximately 0.15 °C.

The melting curve instrument used in this experiment achieves a temperature resolution of ±0.02 °C, allowing for the detection of single-base differences and enabling the identification of single-nucleotide polymorphisms (SNPs). This high level of precision is why the technique is referred to as high-resolution melting (HRM) technology.

In HRM analysis, if the melting temperature difference between two peaks exceeds 0.15 °C, the sequences are considered distinct.

If the Tm difference is less than 0.15 °C, the sequences are considered identical.

### 4.1. Advantages of HRM Technology in Forensic Entomology for Species Identification

From the perspective of this study, adult necrophilous flies are relatively large and easy to detect during sample collection. Individual specimens are less likely to be contaminated during DNA extraction, making molecular-level analysis relatively straightforward [9,50]. Additionally, the mitochondrial genes of flies are highly variable, and the mitochondrial cytochrome oxidase (COX) gene has been widely recognized as a suitable DNA barcoding region for flies [13,51]. By designing experiments based on the nucleotide polymorphisms in these regions, HRM can achieve the species identification of necrophilous flies.

However, in practical forensic applications, higher demands are often placed on the species identification of necrophilous flies. In many crime scenes, adult or larval necrophilous flies found on corpses are often already dead and decomposed. Decomposition inevitably leads to DNA degradation, making it difficult to sequence long DNA fragments [52,53]. In this study, the designed primers targeted a gene region of only 97 bp, and even samples stored for two years yielded reliable detection results. This suggests that our method can successfully identify species in necrophilous insect samples stored for several years.

Since flies undergo complete metamorphosis, the larvae found on corpses are small and morphologically similar, making morphological classification challenging. However, their DNA remains constant throughout their development. Even in the larval stage, species identification can be achieved through molecular DNA analysis [54]. In this experiment, a characteristic peak was successfully obtained for a flesh fly larva, with a Tm value differing by only 0.1 °C from that of *P. dux*, strongly suggesting they belong to the same species. Therefore, DNA-based species identification offers significant advantages over morphological methods in forensic entomology.

In this study, the entire process from DNA extraction to PCR-HRM detection took only 2–3 h, making rapid detection one of its most notable advantages. Compared to sequencing technologies, which are not yet widely accessible and involve lengthy sample submission and sequencing times, HRM analysis is performed directly in the same microcentrifuge tube after PCR amplification. This approach is cost-effective, high-throughput, and fast, while also enabling closed-tube operations to avoid contamination. HRM only detects changes in the fluorescence intensity in PCR products without damaging the DNA, allowing for a further analysis of the PCR products after HRM [18]. This is particularly important for sample preservation in forensic practice.

### 4.2. Current Challenges in Applying HRM

The application of HRM technology in forensic entomology still faces significant challenges. A systematic methodology for HRM-based species identification in forensic entomology needs to be established, including DNA extraction methods, primer selection, PCR-HRM amplification systems, and result interpretation.

#### 4.2.1. Impact of DNA Extraction Methods

This study used a commercial kit for DNA extraction from necrophilous fly samples. However, the degree of DNA digestion and lysis significantly affected the concentration and purity of the extracted DNA. The kit relies on proteinase K for DNA lysis, so the fly specimens must be thoroughly minced. Dried samples are easier to grind and mince, while fresh samples require longer digestion and lysis times. For fresh samples, liquid nitrogen grinding can be used to rapidly dry the tissue, or the samples can be air-dried for 24 h before digestion. The completeness of digestion directly affects the final DNA yield. Additionally, after drying, the mass of fly tissues varies significantly, so fresh and dried samples should not be compared based on the same tissue mass. For fresh samples, it is recommended that they are air-dried for 24 h before extraction, each specimen is weighed, and the thorax and abdomen should be prioritized for larger specimens. Alternatively, leg tissue can be used for DNA extraction, as it is easier to digest and quantify [42,55].

#### 4.2.2. Primer Selection

The primers selected in this study were based on the following three criteria:

HRM principles: Shorter amplification fragments result in a higher accuracy and lower mismatch rates [56]; as such, the product size should ideally be less than 300 bp.

Species identification requirements: DNA barcoding regions specific to necrophilous flies are ideal.

DNA barcoding refers to a short, standardized, and highly variable DNA fragment that can represent a species and is easy to amplify. Therefore, primers should ensure both fly specificity and the presence of variable regions to differentiate between species based on melting temperature differences. The mitochondrial COX gene is widely recognized as a suitable DNA barcoding region for insects, and primers targeting this gene can enable the identification of multiple fly species [51]. For forensic applications, shorter target genes are suitable for long-term sample identification, while longer target genes can be used for fresh samples to improve accuracy and species coverage [57]. Future studies could also explore primers targeting nuclear DNA, as nuclear DNA degrades more slowly than mitochondrial DNA and is more abundant [58]. The ideal primers for the HRM-based classification of necrophilous flies should be species-specific DNA barcoding regions that can amplify multiple fly species and contain interspecies variable sites, enabling systematic classification.

#### 4.2.3. PCR-HRM Amplification System

The PCR amplification system includes factors such as reagent concentrations and amplification conditions. For a 20 µL reaction, the total DNA concentration should be at least 10 ng, as recommended by the PCR mix instructions. Primer concentrations should not be too high, with a final concentration between 0.2 and 0.4 µM. Excessive primer concentrations can lead to primer–dimer formation, which produces characteristic melting curve peaks, typically at lower temperatures [59]. Additionally, a saturation dye must be added to the amplification system for HRM analysis. As dsDNA denatures during heating, the saturation dye binds to the DNA; changes in fluorescence intensity are detected in real time to generate melting curves. This study used Eva Green as the saturation dye, but other dyes such as LC Green, LC Green Plus, SYTO, and ResoLight are also commonly used. The sensitivity of different dyes may vary, and they should be used in conjunction with the appropriate instrument model.

#### 4.2.4. Consistency in HRM Results

After addressing potential issues in the previous steps, HRM can yield reliable results. It is essential to include blank controls in PCR and HRM experiments, especially when multiple peaks are observed in the melting curve, in order to rule out contamination. If contamination is excluded, multiple peaks may indicate the presence of multiple polymorphic regions. However, the exact cause of multiple peaks in melting curves remains unclear, and resolving this issue would represent a significant breakthrough in species identification and melting curve analysis [60].

For species identification, this study found that the Tm values of melting curves for the same species were consistent within the same group of PCR experiments. However, slight variations in peak values were observed when experiments were conducted in separate groups. In future studies, it may be beneficial to include a species or group of species as a ladder in the system to serve as a reference standard.

Additionally, the methodological limitations of molecular biology approaches: heavy reliance on existing databases, particularly for understudied species, poses significant challenges for both primer design and sequence alignment. Taking “*S. princeps*” as an example, we were unable to obtain complete mitochondrial sequences from available databases. Our search revealed only seven partial mitochondrial COXⅠ fragments for this species in NCBI, leaving us without reliable references for primer design. Furthermore, the COXII sequences used in our HRM experiments could not be properly aligned due to insufficient database references.

### 4.3. Analysis of Two PCR-HRM-Based Species Identification Systems in Forensic Entomology

The two primer pairs selected in this study were tested repeatedly to ensure stable amplification and consistent melting curve positions. However, the C1-J-2495/C1-N-2800 primers identified fewer fly species, possibly due to the longer amplification fragment. The ten known standard species used in this study were stored for an extended period, leading to DNA degradation. To extend the storage time of fly samples, they can be preserved in 75% ethanol or stored at −20 °C [61].

After sequencing the PCR-HRM products amplified by the COX2-519/COX2-615 primers, it was evident that different species exhibited DNA polymorphisms in this region, both in terms of sequence and length. However, there were also common conserved regions. When comparing the sequences with the NCBI database, the DNA barcode showed good consistency within the same species, although a 1 bp difference was observed, possibly due to sample degradation or PCR mismatches. Additionally, a phylogenetic tree was constructed for the ten sequenced species (Figure 11), which showed that the species in this region did not exhibit strong taxonomic relationships and were relatively independent. Therefore, this region is not suitable for higher-level (family or genus) classification.

During the repetitive experiments, the five flesh fly species showed slight numerical differences compared to another dataset, but the differences did not exceed 0.15 °C, and the trends were consistent with other data. For unknown samples, the ten known necrophilous fly species must be used as standards, and PCR amplification should be performed simultaneously. If the melting temperature of an unknown sample matches or differs by less than 0.15 °C from a standard species, it is likely to be the same species. We theoretically assumed a Tm difference threshold of 0.15 °C. However, in the HRM analysis of COX2-519/COX2-615, we observed that *H. ligurriens* and *B. peregrina* showed a mere 0.03 °C difference in Tm values. Since DNA melting temperature is determined by multiple contributing factors, including nucleotide sequence composition and GC content, etc., developing more accurate methods for Tm prediction would significantly advance HRM technology.

### 4.4. Limitations and Future Development of PCR-HRM Technology for Necrophilous Fly Species Identification

In the current study, primers were designed specifically for 10 dominant local species, and whether this primer system is applicable to other regions remains unknown. In future experiments, we plan to expand the range of target species and validate the method’s efficacy and potential variations across different geographical areas.

While the technique has demonstrated promising results in identifying degraded samples, its practical application requires the establishment of a standardized identification system. This system should encompass standardized protocols from sample preservation to DNA extraction and species identification. We propose incorporating reference standards to ensure reliability in real-world applications. Specifically, establishing standardized melting temperature (Tm) differentials between target species and reference samples could provide a viable pathway from experimental validation to practical implementation.

## 5. Conclusions

This study preliminarily established a PCR-HRM method for the species identification of necrophilous flies. Using the C1-J-2495/C1-N-2800 primers obtained from the literature, the identification of six necrophilous fly species was achieved. Additionally, the newly designed COX2-519/COX2-615 primers enabled the identification of ten necrophilous fly species, including larval species. These primers demonstrate the potential of DNA barcodes to identify necrophilous flies.

## Figures and Tables

**Figure 1 insects-16-00590-f001:**
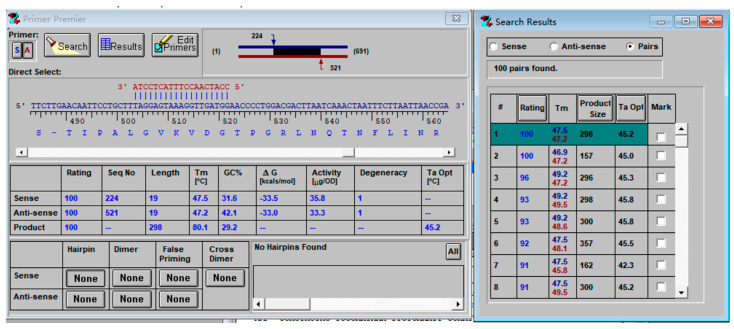
Primer design with the *P. misera* COXII gene sequence.

**Figure 2 insects-16-00590-f002:**
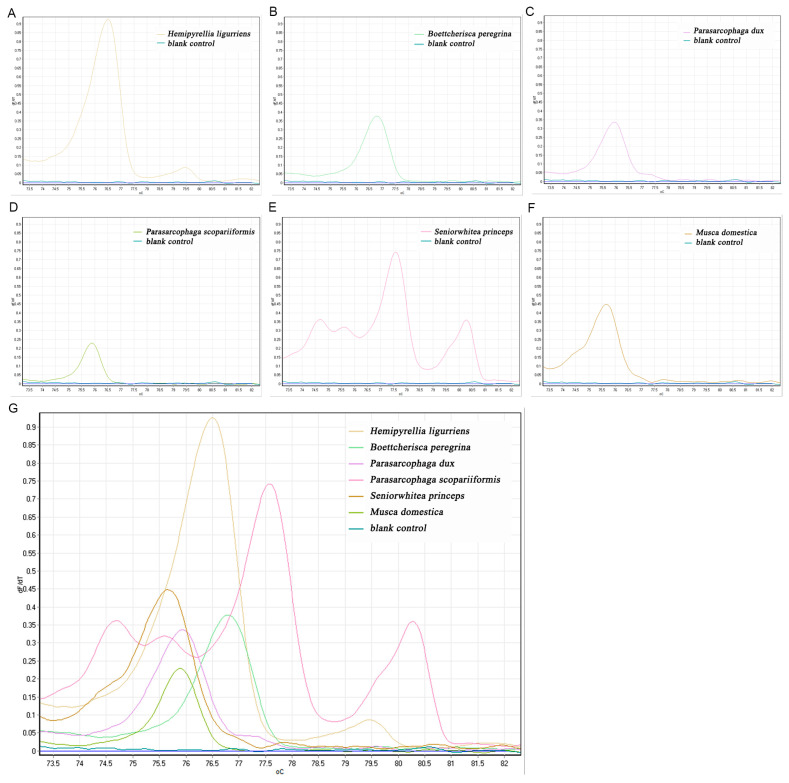
C1-J-2495/C1-N-2800 primers for necrophilic fly HRM species identification. (**A**) *H. ligurriens*; (**B**) *B. peregrina*; (**C**) *P. dux*; (**D**) *P. scopariiformis*; (**E**) *S. princeps*; (**F**) *M. domestica*; (**G**) total melting curve.

**Figure 3 insects-16-00590-f003:**
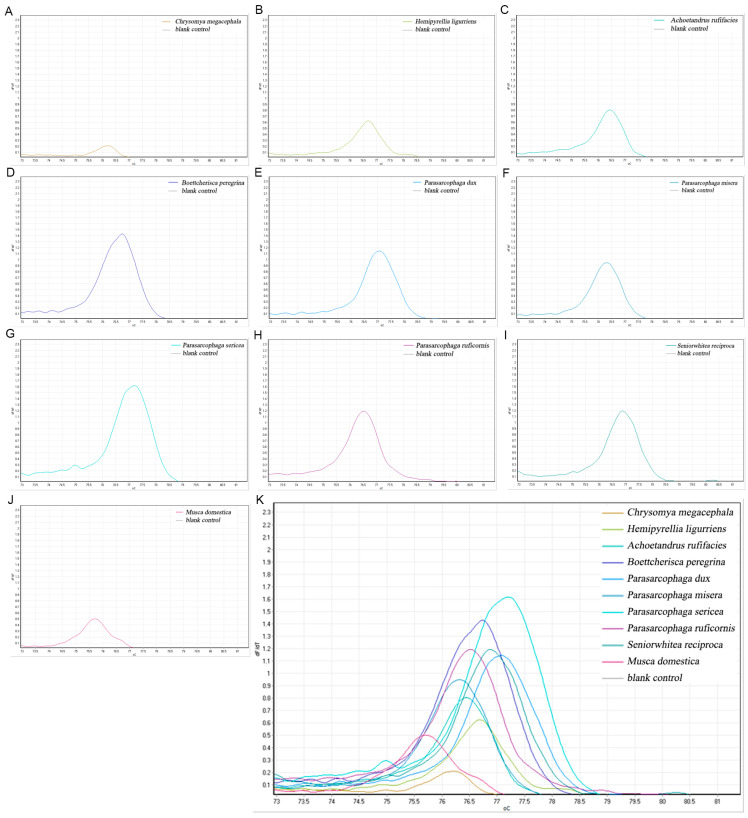
COX2-519/COX2-615 primers for necrophilic fly HRM species identification. (**A**) *C. megacephala*; (**B**) *H. ligurriens*; (**C**) *A. rufifacies*; (**D**) *B. peregrina*; (**E**) *P. dux*; (**F**) *P. misera*; (**G**) *P. sericea*; (**H**) *P. ruficornis*; (**I**) *S. princeps*; (**J**) *M. domestica*; (**K**) total melting curve.

**Figure 4 insects-16-00590-f004:**
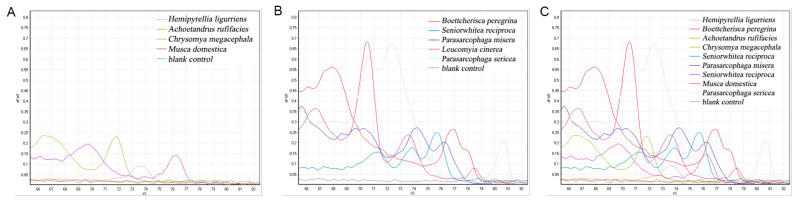
COX2-365/COX2-521 primer HRM results of the Sarcophagidae family showed multiple peaks. (**A**) Single-peak melting curve; (**B**) multi-peak melting curve; (**C**) total melting curve.

**Figure 5 insects-16-00590-f005:**
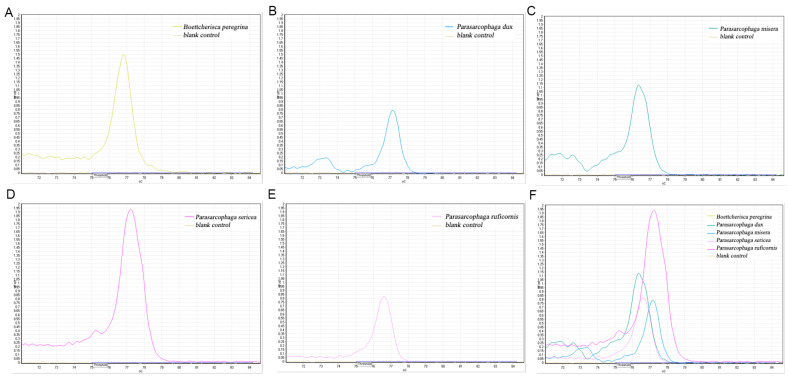
Validation of COX2-519/COX2-615 primers for identification accuracy in five species of Parasarcophaga genus. (**A**) *B. peregrina*; (**B**) *P. dux*; (**C**) *P. misera*; (**D**) *P. sericea*; (**E**) *P. ruficornis*; (**F**) total melting curve.

**Figure 6 insects-16-00590-f006:**
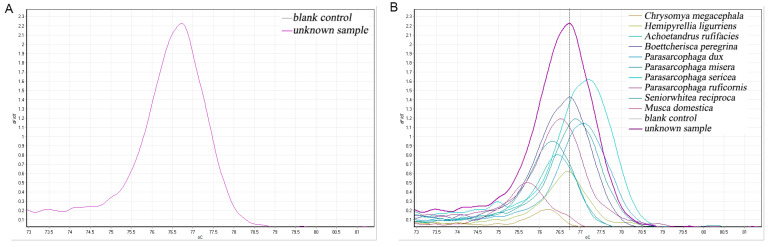
Identification of unknown flies, validating the COX2-519/COX2-615 primers. (**A**) Melting curves of unknown flies. (**B**) Comparison of unknown flies and ten other known samples.

**Figure 7 insects-16-00590-f007:**
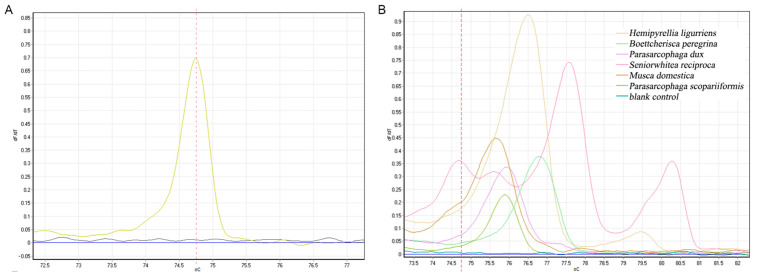
Identification of unknown flies, validating the C1-J-2495/C1-N-2800 primers. (**A**) Unknown fly melting curves. (**B**) C1-J-2495/C1-N-2800 HRM identifies flies.

**Figure 8 insects-16-00590-f008:**
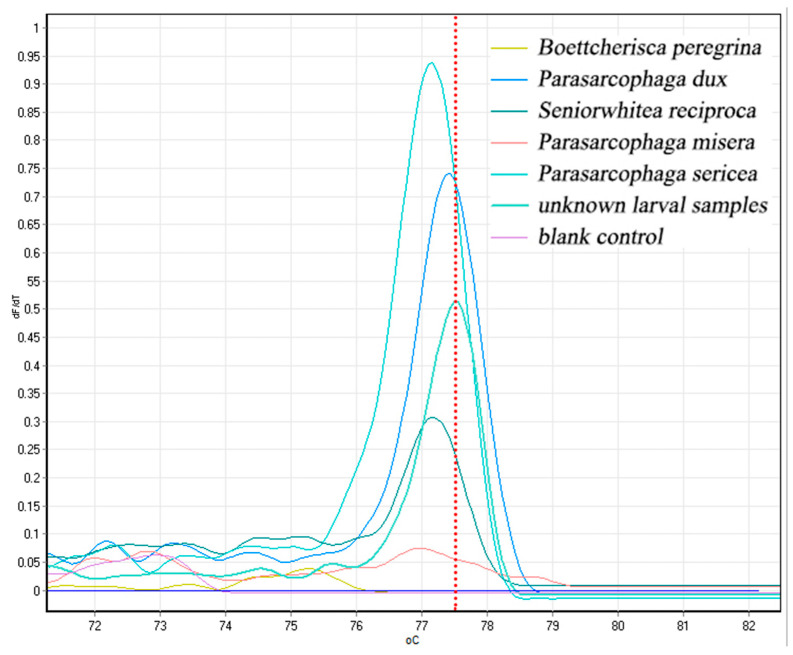
Validation of HRM for necrophilic fly species identification by unknown larval samples.

**Figure 9 insects-16-00590-f009:**
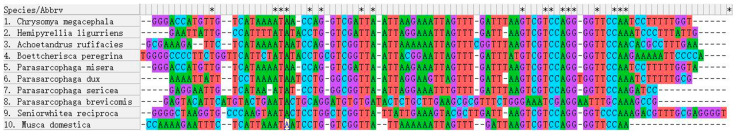
Sequencing results of PCR-HRM products with COX2-519/COX2-615 primers. * The base is identical for all species at that position. Different colors represent distinct bases: A is green, G is purple, C is blue, T is red.

**Figure 10 insects-16-00590-f010:**
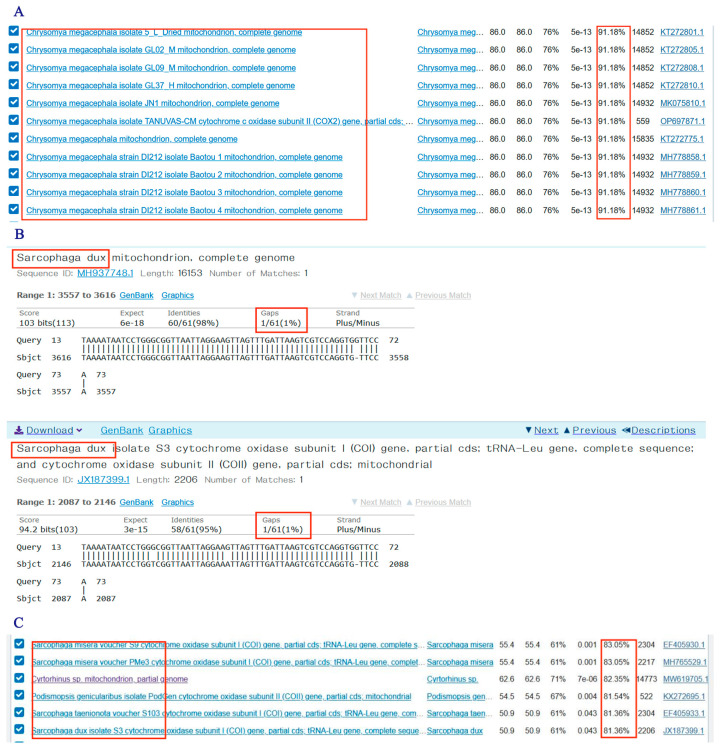
Sequencing results of COX2-519/COX2-615 PCR-HRM in NCBI. (**A**) *C. megacephala*; (**B**) *P. dux*; (**C**) *S. princeps*.

**Figure 11 insects-16-00590-f011:**
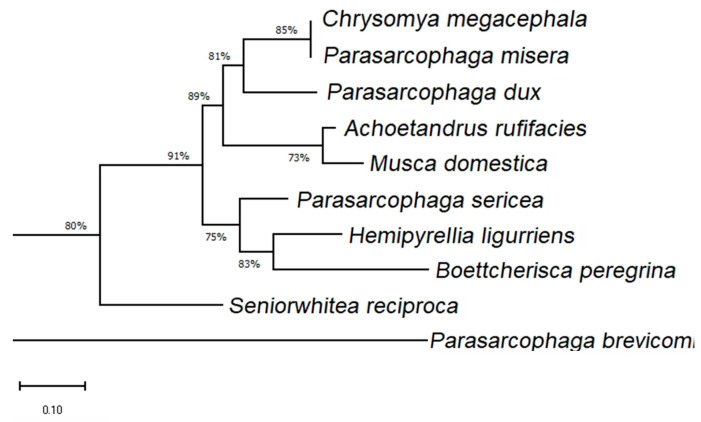
Phylogenetic tree of 10 species within the COX2-519/COX2-615 primer region.

**Table 1 insects-16-00590-t001:** Sequence numbers and positions in GenBank of eight necrophilic fly species; COXⅠ and COXII.

Species	COXⅠ	COXII
*C. megacephala*	NC_019633.1:1428-2961	NC_019633.1:3033-3720
*A. rufifacies*	NC_019634.1:1415-2948	NC_019634.1:3020-3707
*H. ligurriens*	NC_019638.1:1417-2950	NC_019638.1:3022-3709
*P. dux*	NC_039826.1:1411-2944	NC_039826.1:3018-3705
*P. misera*	NC_036107.1:1415-2945	NC_036107.1:3018-3708
*P. peregrina*	NC_023532.1:1413-2946	NC_023532.1:3018-3705
*S. princeps*	NC_042759.1:1434-2967	NC_042759.1:3041-3728
*M. domestica*	NC_024855.1:1410-2943	NC_024855.1:3014-3698

**Table 2 insects-16-00590-t002:** PCR-HRM program.

	Step	Temperature	Time	
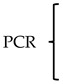	Initial Denaturation	95 °C	10 min	
Denaturation	95 °C	30 s	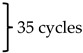
Annealing	Tm	30 s
Extension	72 °C	45 s
	Final Extension	72 °C	10 min	
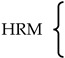	Denaturation	40 °C	1 min	
Cooling	65 °C to 90 °C	1 min	
Melting Curve	40 °C	0.1 °C/s	

**Table 3 insects-16-00590-t003:** PCR-HRM reaction system (20 µL).

Reagent	Volume (µL)
MonAmpTM 2 × Taq mix Pro	10
primer1 (10 µM/dL)	1
primer2 (10 µM/dL)	1
20 × Eva Green	1
ddH_2_O	5
template DNA	2

**Table 4 insects-16-00590-t004:** Concentrations of ten species of DNA were determined after extraction.

Species	*C. megacephala*	*H. ligurriens*	*A. rufifacies*	*B. peregrina*	*P. dux*
DNA Concentration	37.1 ng/µL	7.89 ng/µL	10.87 ng/µL	12.57 ng/µL	138 ng/µL
Species	*P. misera*	*P. sericea*	*P. ruficornis*	*S. princeps*	*M. domestica*
DNA Concentration	73.87 ng/µL	103.67 ng/µL	3.83 ng/µL	51.6 ng/µL	232 ng/µL

**Table 5 insects-16-00590-t005:** Primer list.

Primer Name	Primer Sequence (5′→3′)	Target Gene	Primer Source
C1-J-2495	F:CAGCTACTTTATGAGCTTTAGG	278	[46]
C1-N-2800	R:CATTTCAAGCTGTGTAAGCATC		
16SrF	F:CGCTGTTATCCCTAAGGTAA	289	[47]
16SrR	R:CTGGTATGAAAGGTTTGACG		
COI-1502	F:AGCGGGAATAGTGGGAA	257	newly designed
COI-1758	R:ATGTTAGAGCAGGAGGT		
COI-269	F:CCCCTGCACTAACTTTAC	110	newly designed
COI-378	R:TGATGCTCCACCATGT		
COI-1499	F:TTGAGCGGGAATAGTGG	259	newly designed
COI-1757	R:TGTTAGAGCAGGAGGTA		
COI-359	F:ATTGCCCATGGAGGT	266	newly designed
COI-624	R:CCTGCAGGGTCAAAAA		
COI-59	F:TCGGAGCATGATCTGGTAT	259	newly designed
COI-317	R:GCAGGAGGTAATAATCAAAA		
COI-2600	F:TGTTCATTGATTCCCTCT	184	newly designed
COI-2783	R:AGAAATTACATTTCAAGCTGTGTAA		
COI-118	F:GGTCATCCAGGAGCACTAA	275	newly designed
COI-392	R:GATAACGGAGGGTAAACAG		
COI-794	F:CATTTGGTTCATTAGGGATA	493	newly designed
COI-1286	R:AAGAAGTGTTGAGGGAAGA		
COX2-229	F:GCTTTCCCTTCTTTACGAT	252	newly designed
COX2-479	R:ATTACATCAGCAGCGGTTA		
COX2-224	F:TTATTGCTTTTCCTTCACT	298	newly designed
COX2-521	R:CCATCAACCTTTACTCCTA		
COX2-365	F:TTATTGCTTTTCCTTCACT	157	newly designed
COX2-521	R:CCATCAACCTTTACTCCTA		
COX2-519	F:TGGAACCCCTGGACGACTTA	97	newly designed
COX2-615	R:ACTGTGATTAGCTCCGCAAA		

**Table 6 insects-16-00590-t006:** Primers for the identification of necrophilous fly species.

Primer	Primer Sequence (5′→3′)	Target Gene	Length	Tm (℃)	Source
C1-J-2495	F:CAGCTACTTTATGAGCTTTAGG	COXⅠ	278 bp	48	Literature
C1-N-2800	R:CATTTCAAGCTGTGTAAGCATC				
COX2-519	F:TGGAACCCCTGGACGACTTA	COXII	97 bp	53	Newly designed
COX2-615	R:ACTGTGATTAGCTCCGCAAA				

## Data Availability

The original contributions presented in this study are included in the article. Further inquiries can be directed to the corresponding authors.

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
