# Peer review of "Exploratory Study on DNA Barcode Combined with PCR-HRM Technology for Rapid and Accurate Identification of Necrophilous Fly Species"

_insects, 2025, doi:10.3390/insects16060590_

Round 1

Reviewer 1 Report

Comments and Suggestions for Authors

The study’s goals are relevant and important to forensic entomology, where fast and accurate species identification, especially from degraded or larval material, is a critical component in estimating the postmortem interval (PMI). The manuscript is generally well written, with a logical flow and detailed methodology. However, several aspects of the manuscript require clarification, expansion, or revision to improve its scientific rigor, reproducibility, and overall clarity.

The application of PCR-HRM using a newly designed COX2 primer for species identification of necrophagous flies is a valuable and timely contribution to forensic entomology, particularly for regions with limited access to sequencing infrastructure.

Suggested improvement: Integrate more recent references (post-2020) about HRM in applied forensic contexts, if available.

The systematic primer design, experimental replication, and comparison to known primer sets are strengths. The use of long-stored and larval samples increases the practical applicability of the study.

Sequencing validation and statistical robustness (e.g., Tm differences and HRM reproducibility) support the conclusions.

The manuscript refers to ten species as part of the validation set, but the results for Parasarcophaga scopariiformis are inconsistently reported (e.g., mentioned in HRM results but not in the sample list). Please clarify whether this species was truly part of the tested panel.

Use either necrophilous or necrophagous consistently throughout. Currently, both terms are used interchangeably.

In the HRM results using COX2-519/COX2-615, it is stated that Hemipyrellia ligurriens and Boettcherisca peregrina differ by only 0.03°C in Tm. Given the detection limit of ±0.02°C, please elaborate on whether this close proximity could cause misidentification in practical applications, especially in mixed or degraded samples.

In the BLAST analysis section, the authors note that Seniorwhitea princeps showed <90% identity due to limited database availability. Please acknowledge the implications of limited sequence availability on HRM validation, particularly for less-studied species.

The discussion would benefit from briefly addressing how this HRM protocol could be expanded for field application or integrated with portable forensic technologies 

Consider discussing the potential for multiplexing or broader taxonomic coverage using similar short COX2 fragments.

Only 10 species were tested, yet there are dozens of forensic-relevant Diptera. Authors should acknowledge this limitation, propose future work to expand species panels, and discuss whether the method could be extended to broader geographic ranges.

Line 54: “inferring the time of a crime” consider rephrasing to “estimating time since death, which can aid in narrowing investigative timelines.”

Line 129: space needed after sericea

Delete extra bracket on Line 131 and 132

Line 132: Delete bracket before Musca. Space needed after domestica

Phrases like “successfully amplified all 10 species with a peak” could be clarified to “yielded single, species-specific melt peaks for all 10 species.”

Reviewer 2 Report

Comments and Suggestions for Authors

line 21: no comma at end

line 26: more cost-effective than what? microscopy is cheaper. pls explain.

line 55: pls add also a full review paper to quote [1]→ https://www.researchgate.net/publication/11883594_A_brief_history_of_forensic_entomology

line 57: this is a chinese language paper; many people cannot read chinese. pls add → https://pubmed.ncbi.nlm.nih.gov/15364387/

line 122/123: please add the references to the two fly determination keys (books? journal articles?)

line 149: why only eight species (you mentioned more species in lines 126—132)? 

line 152, 309, 311: latin species name in italics

line 201: do not hyphenate the latin names, just put second part of name below first part of name

line 241f., 254 f., 270 f., 284 f., 295 f., 304 f.: all text inside of the images and at the axes is far too small. please redo it.

figure 10: i cannot read the text in the figures

figure 11: maybe latin names in italics?

Round 2

Reviewer 2 Report

Comments and Suggestions for Authors

line 26: faster (2-3 hours) SPACE and more

line 39:  published studies, which were used as controls; and employed → is the punctuation correct?

line 82 ff: In recent years, with the rapid development of molecular biology, particularly the advancement of species classification technologies based on the analysis of genetic material such as DNA, molecular species identification techniques have begun to play an increasingly important role in forensic entomology [9-11]. → this is no so recent (2008/2011)

lines 191; 230 ff. → the table (n my .pdf version) is cut into two pieces

line 208: their DNA was re-extracted the DNA, → grammar 

line 480: Additionally, We mustwe

i like the many images 🫱🏽‍🫲🏾
